METHODS

# Inferring pathway activity from single-cell and spatial transcriptomics data with PaaSc

**Xiqi Liao**[1☯], **Yuyang Hong**[1☯], **Yan Feng**[1☯], **Henghui Li**[1], **Hai Fang**[2], **Jiantao Shi**[1*]

1 Key Laboratory of RNA Innovation, Science and Engineering, Shanghai Institute of Biochemistry and Cell Biology, Center for Excellence in Molecular Cell Science, Chinese Academy of Sciences, Shanghai, China, 2 Shanghai Institute of Hematology, State Key Laboratory of Medical Genomics, National Research Center for Translational Medicine at Shanghai, Ruijin Hospital, Shanghai Jiao Tong University School of Medicine, Shanghai, China

☯ These authors contributed equally.
* jtshi@sibcb.ac.cn

## Abstract

Recent advances in single-cell and spatial transcriptomics have revolutionized our understanding of cellular heterogeneity. However, translating high-dimensional data into functional pathway insights remains challenging. To address this obstacle, we developed PaaSc (Pathway activity analysis of Single-cell), a computational method for inferring pathway activity at single-cell resolution. PaaSc employs multiple correspondence analysis to simultaneously project cells and genes into a common latent space and selects pathway-associated dimensions through linear regression to infer pathway activity scores. We validated PaaSc across diverse benchmarking datasets, including those that jointly profiled protein and RNA levels, as well as large-scale cancer scRNA-seq cohorts. Compared with state-of-the-art methods, PaaSc demonstrated superior performance in multiple applications: scoring cell type-specific gene sets, identifying cell senescence-associated pathways, and exploring GWAS trait-associated cell types. Importantly, PaaSc maintained accuracy despite batch effects and demonstrated robust performance across different data modalities, including scATAC-seq and spatial transcriptomics data. Our results demonstrate that PaaSc accurately captures dynamic cellular states and spatial patterns, thereby advancing our understanding of cellular dynamics, aging, and disease mechanisms.

## Author summary

Understanding how individual cells function and communicate is crucial for advancing medicine and biology. Recent technologies allow scientists to study thousands of cells individually, revealing that even cells of the same type can behave very differently from one another. However, making sense of this overwhelming amount of data remains a major challenge, particularly when

**Data availability statement:** All datasets used in this study were obtained from publicly available resources. The PBMC3k, Bmcite, and pancreas scRNA-seq datasets were accessed through the SeuratData R package. Human cortex, pancreas, liver immune cell, and spleen immune cell scRNA-seq datasets were obtained from the 'scRNAseq' R package. Senescence-related scRNA-seq data were downloaded from the Gene Expression Omnibus (GEO) database with accession number GSE175533, GSE102090, GSE119807, and GSE115301. The human PBMC REAP-seq data utilized for benchmarking were obtained from the GEO database under the accession number GSE100501. Single-cell tumor immune data were acquired from the TISCH database (http://tisch.comp-genomics.org/). Single-cell transcriptomics data from 20 mouse tissues were accessed via Figshare (https://figshare.com/articles/dataset/Processed_files_to_use_with_scanpy_/8273102). Mouse brain scRNA-seq data were obtained from the Signac repository (https://signac-objects.s3.amazonaws.com/allen_brain.rds). The processed 10x Genomics Visium mouse brain spatial transcriptomics data were retrieved from Zenodo (https://doi.org/10.5281/zenodo.8201825). The 10x multiomics data were downloaded from the 10x Genomics website (https://cf.10xgenomics.com/samples/cell-arc/1.0.0/pbmc_granulocyte_sorted_10k/pbmc_granulocyte_sorted_10k_filtered_feature_bc_matrix.h5). The processed human NSCLC and DLPFC spatial transcriptome datasets were obtained from Zendo (https://doi.org/10.5281/zenodo.8201825). The source data generated in this study were deposited at Zenodo and can be accessed at https://doi.org/10.5281/zenodo.17138447. The PaaSc software is freely available at https://github.com/yoyoong/PaaSc. The script used in this study was deposited at Zenodo and can be accessed at https://doi.org/10.5281/zenodo.17136774.

**Funding:** This study is supported in part by the National Natural Science Foundation of China (32270691 to JS). The funders had no role in study design, data collection and analysis, decision to publish, or preparation of the manuscript.

**Competing interests:** The authors have declared that no competing interests exist.

attempting to understand which biological processes are active in each cell. We developed a new computational approach called PaaSc to solve this problem. Our method involves analyzing the activity levels of biological pathways—coordinated sets of genes that work together to perform specific cellular functions—in individual cells. Biological pathways can be viewed as different departments in a company: some are responsible for energy production, others manage cell division, and yet others respond to stress. We tested our approach using multiple datasets and reported that it outperforms existing methods across various applications, including identifying cell types, detecting cellular aging, and understanding disease-related processes. Importantly, our method works reliably even when data come from different laboratories or experiments. We believe that this tool will help researchers better understand how cells behave in health and disease, potentially leading to new therapeutic strategies.

## Introduction

Emerging single-cell and spatial transcriptomic approaches have fundamentally changed our ability to investigate cellular complexity and tissue structure, revealing gene expression patterns with exceptional cellular detail [1,2]. While these technologies have generated vast amounts of high-dimensional data, translating this information into meaningful biological insights remains challenging [3], particularly in understanding the functional states of cells through pathway activity analysis [4]. Cellular pathways comprise interconnected molecular networks that govern diverse biological functions, including signal transduction and metabolic regulation [5]. Traditional bulk RNA sequencing approaches have provided valuable insights into pathway activation patterns at the tissue level [6], but these methods mask the inherent heterogeneity present at the single-cell level. Precise measurement of signaling pathway activity at the single-cell level is essential for deciphering how cells function, how tissues are organized, and how diseases develop.

Owing to the initial lack of analytical tools specifically designed for single-cell data, methods originally developed for bulk gene expression analysis were adapted to assess pathway activity at the single-cell level. Notable examples include random walk approaches such as GSVA [7] and ssGSEA [6], which generate sample-level pathway scores by applying a Kolmogorov–Smirnov-like random walk statistic to within-sample gene rankings. In recent years, several computational tools have been developed to infer pathway activities at the single-cell level. AUCell generates cell-level gene set scores on the basis of gene ranks within each cell using the area under the receiver operating characteristic curve (ROC) [8]. VAM employs a variance-adjusted Mahalanobis distance metric to quantify pathway activities by considering both the mean expression and covariance structure of pathway genes within each cell [9]. scGSEA leverages non-negative matrix factorization (NMF) to decompose single-cell data into gene and cell weight matrices, then applies GSEA to gene weights for pathway enrichment scoring, which is combined with cell weights

to quantify single-cell pathway activities [10]. Recently, multiple correspondence analysis (MCA) has been used through CelliD [11] and GSDensity [12] to project cells and genes into a shared latent space and to quantify pathway activity levels across cells. However, the performance of these methods was evaluated only on several selected datasets.

Here, we present PaaSc, a computational framework for inferring pathway activity at the single-cell level. Through this approach, MCA is used to project both cells and genes into a common low-dimensional space, and then biologically relevant components are identified via linear regression analysis. These components are subsequently utilized to compute pathway activity scores by weighting cellular coordinates in the latent space. We systematically validated PaaSc on multiple benchmark datasets, including multimodal datasets with paired RNA and protein measurements, along with large-scale cancer scRNA-seq cohorts. Comparative analyses demonstrated the superior performance of PaaSc over existing state-of-the-art methods, showing enhanced sensitivity and precision across diverse biological applications.

## Results

### An overview of the PaaSc method

The PaaSc workflow calculates pathway activity scores at the single-cell level through four sequential steps. First, cell-gene shared space construction: PaaSc employs multiple correspondence analysis (MCA) to project cells and genes from scRNA-seq data into a shared low-dimensional space. Implemented using the CelliD package [11], this process generates loading and embedding matrices. The resulting MCA representation functions as a biplot where spatial relationships—between cells, between genes, and between cells and genes—reflect their underlying associations (Fig 1A). Second, pathway-relevant dimension identification: Linear regression analysis is applied to the loading matrix to identify dimensions relevant to the pathway of interest. This model evaluates whether loading factors are attributable to the target pathway versus background components. Third, dimension weight calculation: Dimensions showing significant associations ($P < 0.05$) are retained, with their significance quantified using t-statistics—calculated as the ratio of regression coefficients to their standard errors (Fig 1B and 1C). These t-statistics serve as the primary weight for each dimension, while the proportion of variation explained by each dimension provides a secondary weight. Fourth, normalized score generation: Raw pathway activity scores are computed through a weighted sum of the embedding matrix, followed by z-score normalization to standardize the scores (Fig 1D).

The normalized scores can be employed for various downstream analyses, including cell type annotation and testing for cluster and spatial associations. Additionally, the scores can be further binarized to classify cells into positive or negative states, assuming a bimodal distribution of pathway activity (**Fig 1E**). The PaaSc algorithm is further detailed in pseudocode (S1 Fig) and demonstrated using the human PBMC3k dataset to assess B cell activation pathway activity at the single-cell level (S2 Fig).

### Evaluation of the performance of PaaSc for scoring cell type-specific gene sets

To assess the performance of PaaSc, we analyzed RNA expression and protein sequencing (REAP-seq) data from human peripheral blood mononuclear cells (PBMCs) [13]. This dataset enabled reliable cell classification through simultaneous protein marker measurements at the single-cell level, identifying nine distinct cell types: CD4 + T cells, CD8 + T cells, natural killer (NK) cells, plasmacytoid dendritic cells (pDCs), dendritic cells (DCs), CD14 + monocytes, CD16 + monocytes, and megakaryocytes (Mk) (Fig 2A). We performed a comprehensive benchmarking study comparing PaaSc with established pathway activity inference methods using cell type-specific markers (S3A Fig) [11]. The comparisons included single-cell analysis tools (AUCell, CelliD, GSdensity, VAM, and scGSEA) and bulk RNA-seq analysis tools adapted for single-cell applications (GSEA and GSVA). Each tool generates a score for every cell that quantifies the activity of a given cell type-specific gene set. To evaluate this discriminatory ability, we constructed ROC curves using the predefined cell labels as the ground truth. The area under the ROC curve (AUC) served as our primary metric, measuring each tool's ability to distinguish target cell type cells (true positives) from non-target cells (true negatives).

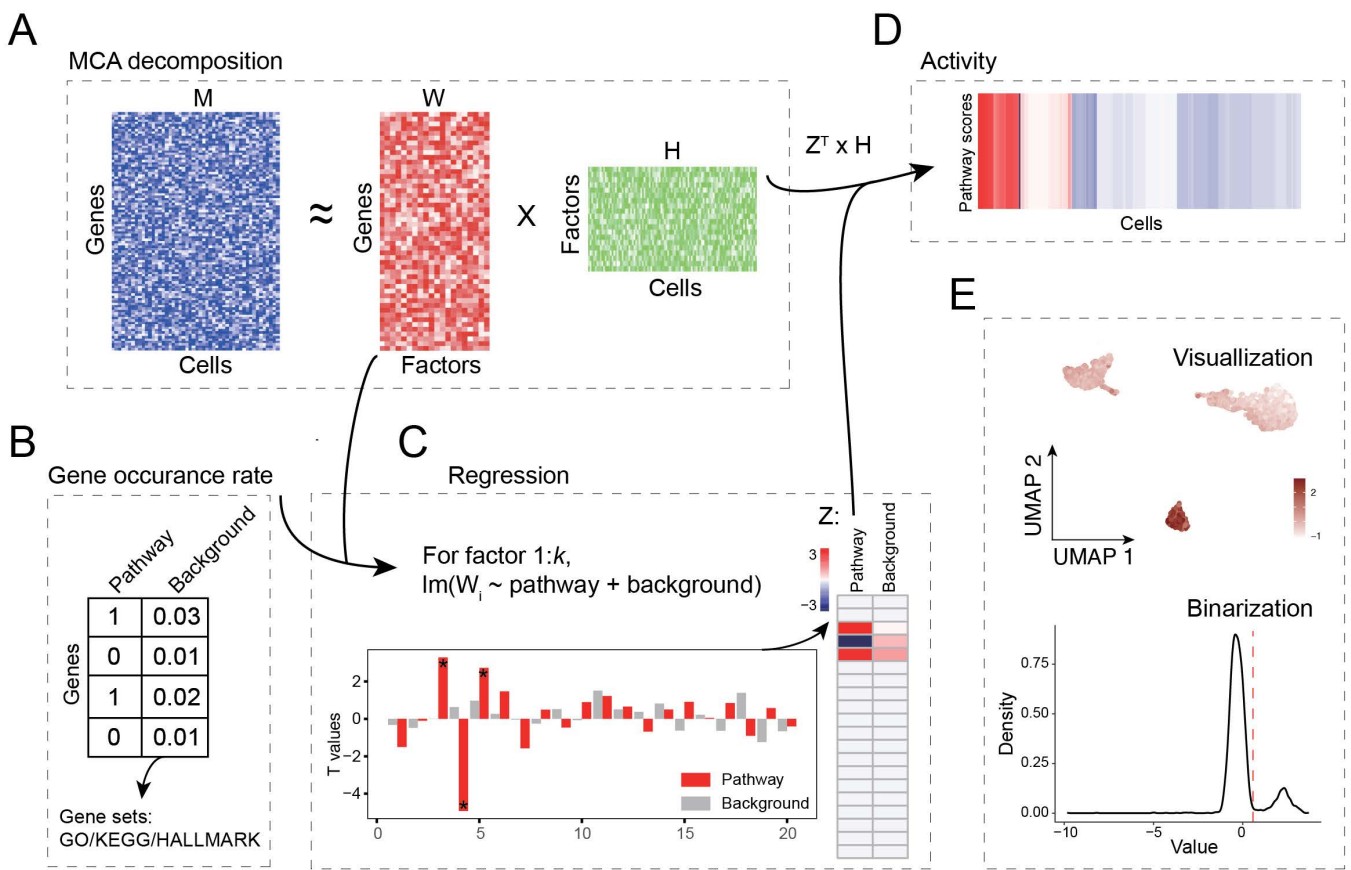

**Fig 1. Overview of the PaaSc approach.** (A) Through the PaaSc approach, multiple correspondence analysis (MCA) is employed to perform dimensionality reduction of the gene expression matrix, projecting both cells and genes into a shared low-dimensional orthogonal coordinate space. The resulting biplot representation captures spatial relationships between cells, genes, and their associations in the reduced-dimensional space. (B) The relative frequencies of genes in the pathway of interest and background gene sets were constructed to assess their contributions to the identified dimensions. (C) Ordinary linear regression was applied to identify significant dimensions associated with the pathway of interest. Dimensions with significant associations ($P < 0.05$) were retained, and their significance levels were quantified using t-statistics, computed as the ratio of regression coefficients to their corresponding standard errors. (D) Raw pathway activity scores were computed as a weighted sum of the embedding matrix, incorporating both t-statistics and the proportion of variation explained as weights. These scores were then z-score normalized for downstream analysis and visualization. (E) Normalized pathway activity scores were used for cell type assignment, cluster association testing, and spatial analysis.

Our comparative analysis revealed that PaaSc and GSdensity achieved the highest performance, with AUC values of approximately 0.99, while CelliD ranked third, with an AUC of 0.96 (Fig 2B). Notably, all three top-performing methods utilize MCA, demonstrating the effectiveness of this dimension reduction technique in single-cell analysis. Unexpectedly, the bulk RNA-seq methods GSEA and GSVA outperformed their single-cell counterpart scGSEA, achieving AUC scores of 0.92 and 0.91, respectively. We next assessed the robustness of these tools by introducing random genes as cell markers, with the proportion varying from 10% to 80%. Our analysis revealed that the performance of all tools declined proportionally as the fraction of random genes increased (Fig 2C). As expected, MCA-based methods demonstrated superior resilience: they not only maintained the highest AUC scores across all noise conditions but also exhibited minimal sensitivity to random gene introduction.

We further evaluated the performance of these methods on a more challenging task: single-cell annotation using gene set scores of cell type markers. In this analysis, each cell was scored against markers from 20 predefined cell types

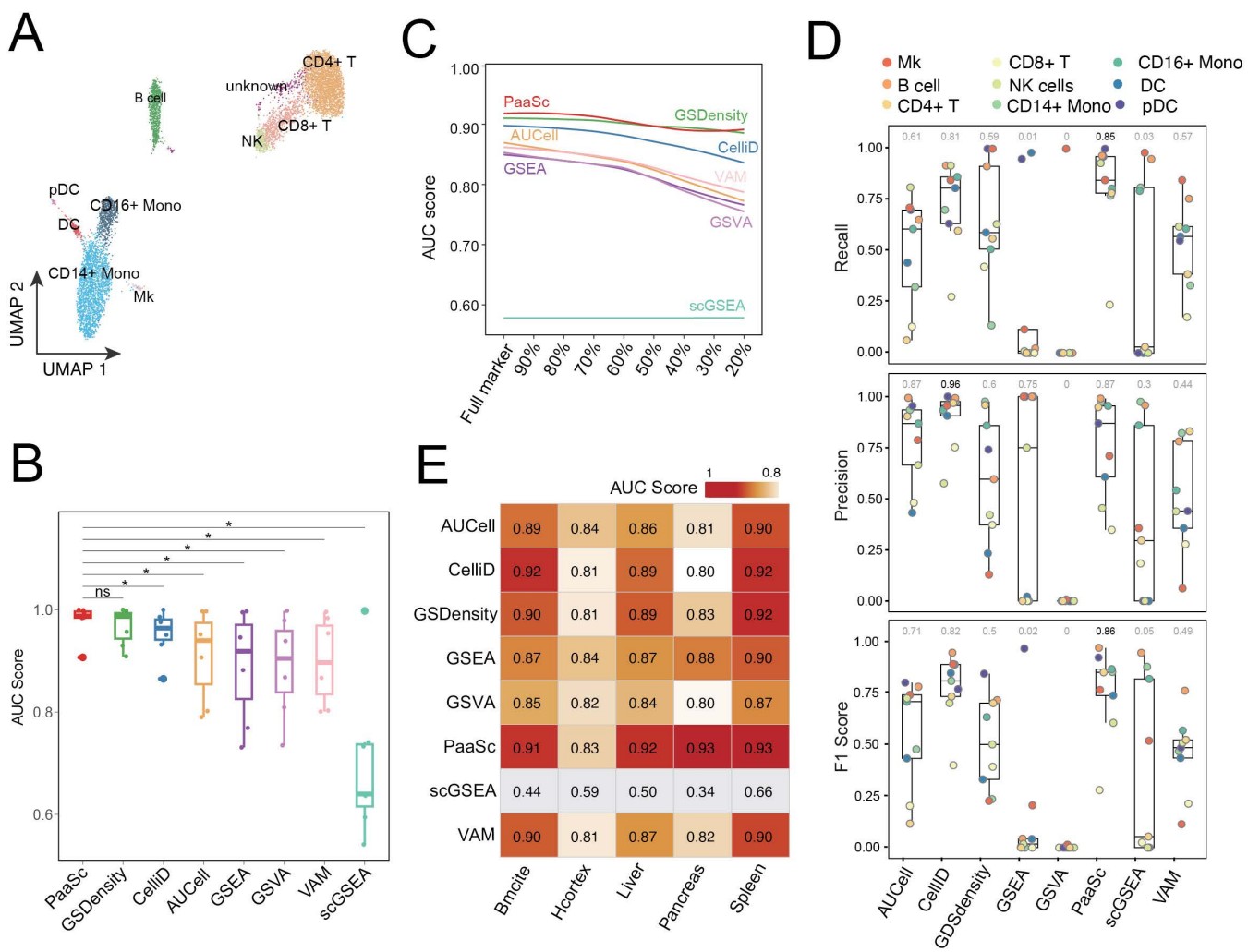

**Fig 2. Comprehensive performance evaluation of PaaSc and other gene set scoring methods using REAP-seq data and benchmark datasets.** (A) UMAP visualization of nine distinct cell populations identified in human PBMCs profiled by REAP-seq, including CD4+T cells, CD8+T cells, natural killer (NK) cells, plasmacytoid dendritic cells (pDCs), dendritic cells (DCs), CD14+monocytes, CD16+monocytes, and megakaryocytes (Mk). (B) Performance comparison of different gene set scoring methods using established cell type-specific markers. Box plots show the distribution of AUC scores across all cell types for each method. The centerline represents the median value, the box limits indicate the first and third quartiles, and the whiskers extend to the minimum and maximum values. (C) Assessment of the robustness of the gene set scoring tools against random noise. The line plot shows the mean AUC scores of different methods when varying proportions (10–80%) of random genes were introduced into cell type-specific marker sets. (D) Cell type annotation performance of different methods evaluated using marker genes from 20 predefined cell types. Box plots show the distributions of recall (upper), precision (middle), and F1 scores (lower) across all cell types. Unassigned cells were excluded from the calculation. (E) Cross-dataset validation using five independent benchmark datasets (Liver, Pancreas, Spleen, Bmcite, and Hcortex). The heatmap shows the mean AUC scores for each method across different datasets.

and assigned to the cell type with the highest score. For multiclass prediction evaluation, we compared recall, precision, and the F1 score, which balances precision and recall. PaaSc achieved the highest median recall rate (0.85) (S3B Fig), followed by CelliD (0.81) (Fig 2D). In terms of precision, PaaSc achieved the second-highest rate of 0.87, while CelliD attained the highest precision of 0.96—this was achieved by designating 20% of cells (n = 2,000) as unassigned. When evaluated using the F1 score, PaaSc outperformed CelliD (0.86 versus 0.82). Although GSdensity performed well in discriminating individual cell types, its effectiveness in comprehensive cell annotation was limited, resulting in an F1 score of

0.5. Notably, traditional bulk RNA-seq methods, such as GSEA and GSVA, exhibited minimal classification capability, with F1 scores approaching zero.

Additionally, we evaluated five benchmark datasets previously used to assess the performance of gene-set scoring tools, including GSdensity [12]. These datasets include Liver [14], Pancreas [15], and Spleen [14], Bmcite, and Hcortex [16]. All methods except scGSEA achieved strong performance, with AUC values higher than 0.8, and PaaSc ranked first in three datasets and second in the remaining two (Fig 2E).

## Benchmarking of PaaSc for large-scale cancer scRNA-seq data

To assess the performance of PaaSc on a larger cohort, we obtained annotated scRNA-seq datasets from the Tumor Immune Single-cell Hub (TISCH) database [17]. Among the 118 datasets with raw UMI counts, 41 contained B cells, 79 contained CD8＋T cells, 92 contained macrophages, 27 contained NK cells, and 11 contained regulatory T cells (Tregs) (Fig 3A and S1 Table). To evaluate the ability of each tool to distinguish target cell types from other populations, we calculated the AUC scores on the basis of cell type-specific markers/pathways. As a case study, we focused on the B cell activation pathway from Gene Ontology (GO), which is known to be active in B cells and was previously utilized to validate GSdensity [12]. Among the 41 datasets containing defined B cell populations, PaaSc ranked first, with a median AUC of 0.97, significantly outperforming CelliD (0.92) and VAM (0.90) (Fig 3B). Notably, PaaSc achieved the highest performance in 29 datasets, while CelliD and VAM led in only 3 datasets (Fig 3C). Furthermore, testing of B cell-specific markers revealed similar results, with PaaSc, scGSEA, and CelliD exhibiting comparable performance and outperforming the other tools (S4 Fig).

We extended our evaluation to four additional immune cell populations: CD8＋T cells, macrophages, Tregs, and NK cells. For each cell type, the datasets were grouped according to which tool achieved the highest performance, as measured by the AUC of the cell type-specific gene sets (Fig 3D and S2 Table). PaaSc achieved the highest AUCs for more datasets across all four datasets. For example, in the analysis of 92 macrophage-containing datasets, PaaSc outperformed competing methods in 36 cases (39%), while scGSEA, despite its marginally higher mean AUC score, led in only 26 cases (28%) (Fig 3E).

## Exploring cell senescence-associated pathways in tumor-infiltrating immune cells using PaaSc

Building upon the ability of PaaSc to accurately quantify single-cell gene-set activity, we investigated its application in analyzing biological aging processes. We focused on a previously established robust cell senescence signature with broad applicability across tissues and species [18]. To independently validate this signature, we analyzed a curated cohort comprising 41 bulk RNA-seq datasets, each containing senescent and non-senescent samples (S3 Table) [19]. Pathway enrichment analysis revealed significant activation (FDR＜0.05) of this signature in the senescent group across 70% (29/41) of the analyzed datasets (Fig 4A). We then analyzed a SMART-seq2 dataset (GSE115301) comparing IMR90 cells treated with 4-hydroxytamoxifen for senescence induction with untreated proliferating controls [20]. PaaSc analysis revealed progressive activation of senescence signatures (Fig 4B), with significant differences between day 4 and day 2 ($p = 8.4 \times 10^{-9}$) and between senescent cells and day 4 ($p = 4.5 \times 10^{-4}$). In contrast, CelliD and GSdensity only detected significant differences between days 4 and 2 and failed to distinguish day 4 from senescent cells ($p = 0.57$ and $0.29$, respectively; S5A Fig).

To further validate these findings, we analyzed scRNA-seq data (GSE175533) [21] from WI-38 cells across sequential population doublings (PDL 20–50). PaaSc-quantified pathway activities showed strong correlation with increasing passage numbers (Pearson $r = 0.7$; S5E Fig). In contrast, neither CelliD nor GSdensity exhibited comparable progressive changes (Pearson $r = 0.38$ and $0.51$, respectively; Fig 4C). To evaluate the discriminative power of PaaSc between senescent and non-senescent cells at the single-cell level, we analyzed a publicly available dataset (GSE102090) containing both proliferating and senescent cell populations [22]. While all three MCA-based tools showed statistically significant

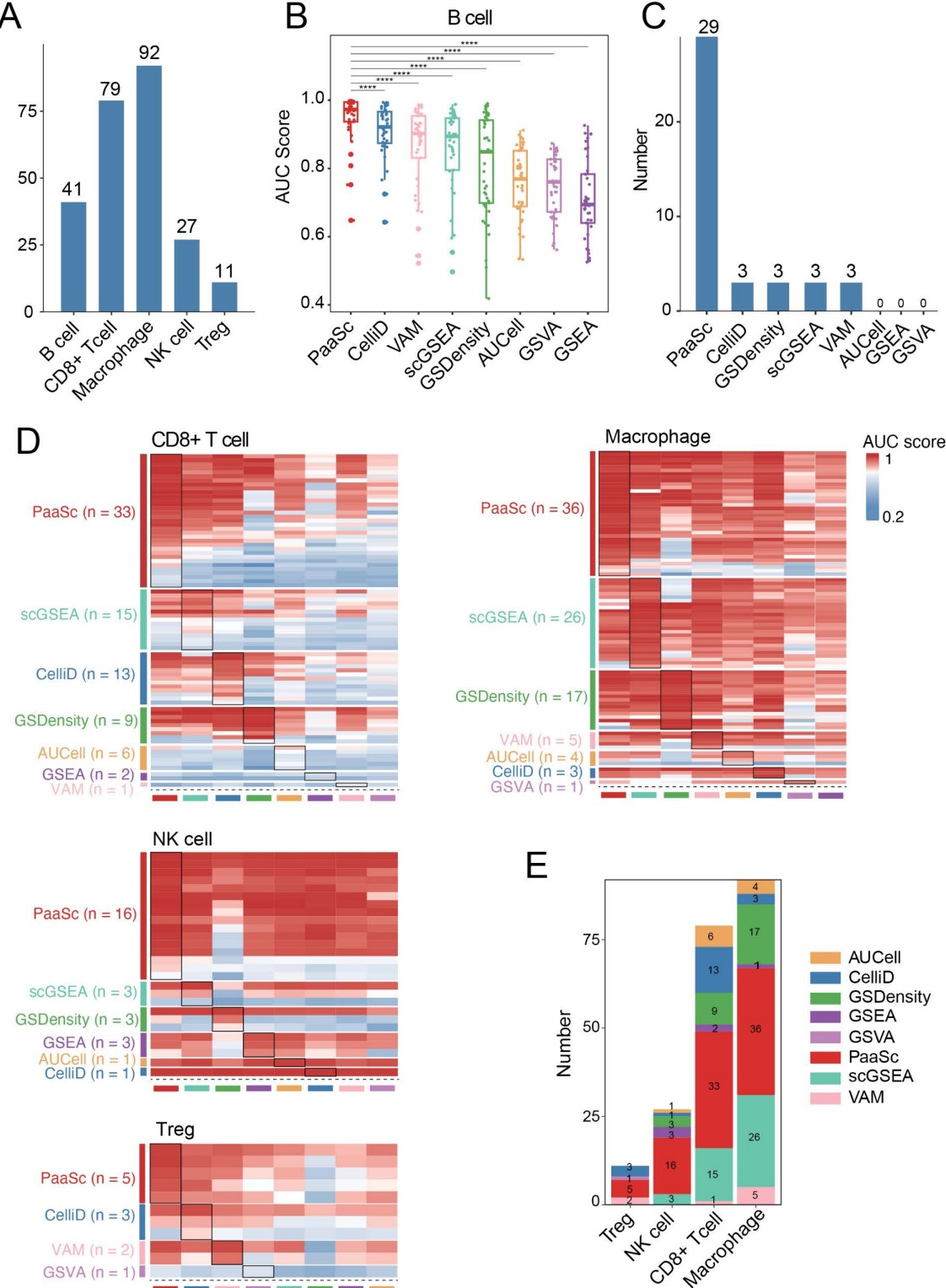

**Fig 3. Evaluation of the performance of PaaSc on 136 annotated scRNA-seq datasets.** (A) A bar plot illustrating the number of datasets containing each of the five cell types analyzed: B cells, CD8+T cells, macrophages, Tregs, and NK cells. (B) Box plots comparing the performance of PaaSc and 7 competing tools in distinguishing B cells from other cell populations, as measured by AUC scores based on B cell-specific markers. (C) A bar plot

showing the number of datasets in which each tool achieved the highest AUC score for B cell identification. (D) Four heatmaps comparing the performance of PaaSc and competing tools in scoring pathway activity for CD8 + T cells, macrophages, Tregs, and NK cells. The datasets were grouped according to which tool achieved the highest accuracy for each cell type. (E) Stacked bar plots summarizing the number of datasets in which each tool achieved the best performance for each cell type.

differences between groups (S5B Fig), PaaSc demonstrated superior discrimination, with an AUC of 0.99, substantially outperforming GSdensity (AUC = 0.63) and CelliD (AUC = 0.61) (Fig 4D, upper panel). These results were further validated using two additional datasets, GSE119807 [23] and GSE115301 [20] (Figs 4D, middle and lower panels, S5C and S5D).

Finally, we searched for pathways that potentially drive cell senescence using PaaSc by testing the association between the activities of the senescence signature and hallmark pathways from MSigDB [5]. We tested 140 datasets from the TISCH database; in total, there were 5 cell types, including CD8 + T cells, CD4 + T cells, B cells, NK cells, and monocytes/macrophages, across 613 patients. As expected, proliferation-associated pathways were negatively correlated with senescence, such as DNA repair, E2F targets, MYC targets. We also found more than 10 pathways that were positively correlated with senescence across all 5 cell types (correlation coefficient > 0.2, FDR < 5%), including coagulation, EMT, KRAS, NF-κB, hypoxia, complement, angiogenesis, IL6 JAK STAT3 signaling, P53, and the TGF pathway (**Fig 4E**). Many of these associations are supported by previous studies [24]. For instance, TNF alpha, the P53 pathway, and IL16 have been identified as downstream signals in T-cell senescence [24]. Additionally, Coagulation factor IX (F9) was identified by CRISPR screening as a regulator of senescence [25], and cellular senescence was found to be associated with age-related blood clots [26]. Furthermore, NF-κB promotes the senescence-associated secretory phenotype (SASP) and enhances chemosensitivity [27]. However, GSDensity showed systematic bias toward positive correlations between senescence signatures and signaling pathways (S5F Fig), which contradicts established biological knowledge.

## Identification of GWAS trait-associated cell types using PaaSc

scRNA-seq has emerged as a powerful tool for identifying disease-associated cell types through polygenic enrichment analysis of trait-associated genes identified by genome-wide association studies (GWASs) [28]. To evaluate the performance of PaaSc in identifying GWAS trait-associated cell types, we analyzed a mouse dataset comprising 19 sorted cell types categorized into blood, brain, or other categories [28]. We first examined the enrichment patterns of lymphocyte count-associated genes. Consistent with previous findings, this signature resulted in higher PaaSc scores for blood lineage cell types, especially for CD4 + T cells, regulatory T cells, and CD8 + T cells (Figs 5A and S6A). The PaaSc scores exhibited a distinct bimodal distribution, enabling the classification of cells into positive and negative states on the basis of lymphocyte count-associated gene enrichment (Fig 5B). Fisher's exact test demonstrated significant enrichment of positive cells across all seven blood cell types but not across all cell types (FDR < 0.05; Fig 5C).

We extended this analytical framework to evaluate 22 diseases and traits spanning blood-related, brain-related, and other categories. For comparative analysis, we defined positive cells as the top 5% of cells with the highest scores for each method tested (S6B Fig). The PaaSc results revealed clear cell type-specific associations: blood/immune cell types correlated with blood/immune-related diseases and traits, brain cell types correlated with brain-related conditions, and other cell types correlated with their corresponding disease/trait categories (Fig 5D, **left panel**). In contrast, CelliD demonstrated limited sensitivity and failed to identify associations between medium spiny neurons (MSNs) and several neuron-related traits, including college education, body mass index (BMI), and smoking status [28]. Furthermore, CelliD failed to detect associations between oligodendrocyte precursor cells (OPCs) and any brain-related diseases and traits (Fig 5D, **middle panel**). GSDensity exhibited notably lower specificity, incorrectly identifying associations between blood/immune cell types and neuron-related traits (Fig 5D, **right panel**).

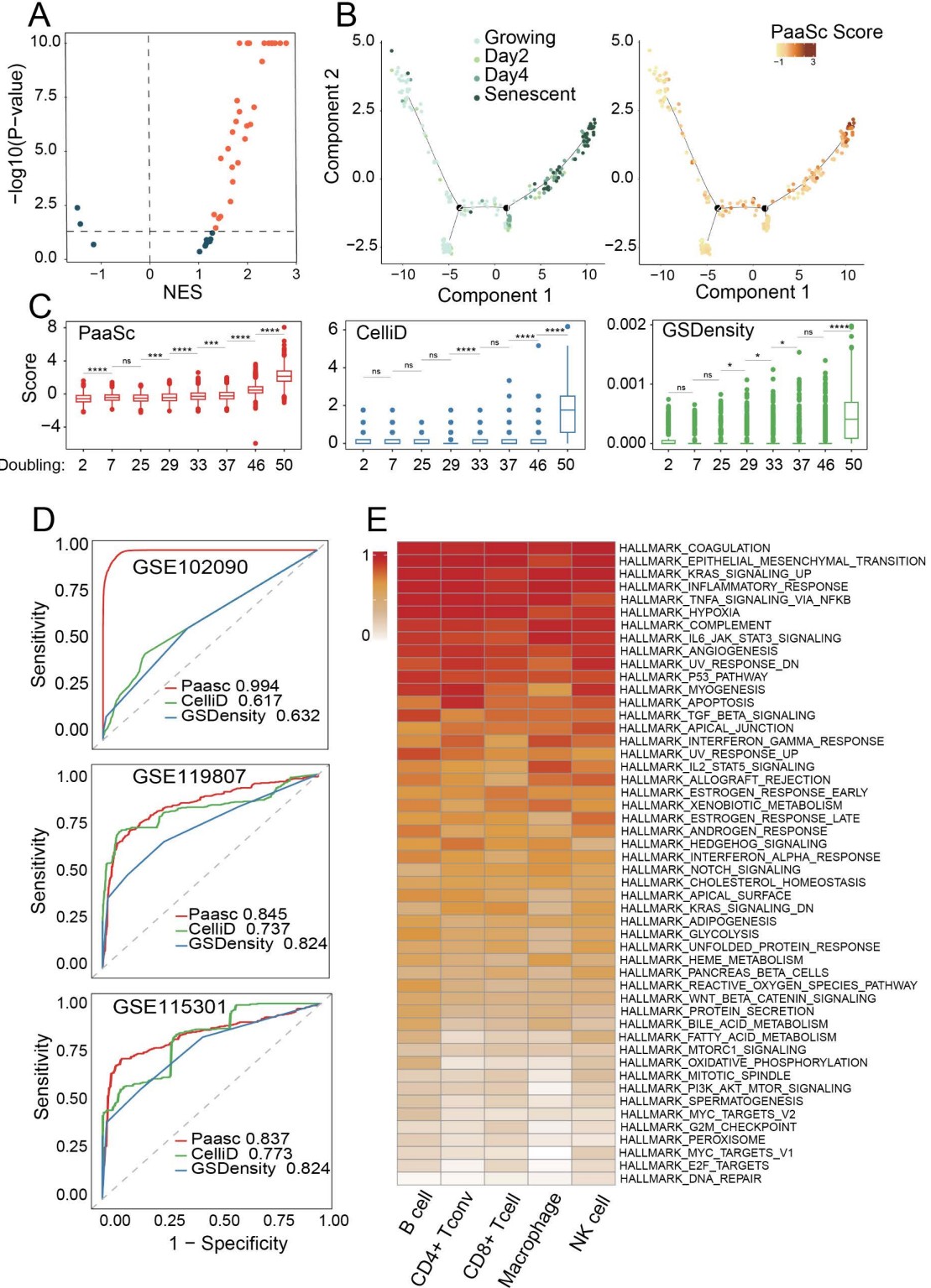

**Fig 4. Application of PaaSc in analyzing biological aging processes.** (A) Enrichment of the senescence signature in the bulk RNA-seq datasets. A volcano plot showing the enrichment of senescence signatures across 50 bulk RNA-seq datasets. The x-axis represents the normalized enrichment score (NES), and the y-axis represents the permutation p-value in -log10 transformation. (B) Assessment of senescence in IMR90 cells treated with

4-hydroxytamoxifen. Cells were projected into two-dimensional space using Monocle2, with labels indicating either time points or calculated PaaSc scores. (C) Assessment of senescence in WI-38 cells across sequential population doublings. Senescence scores were calculated using PaaSc, CelliD, and GSDensity. Statistical significance was assessed by a one-sided Wilcoxon rank sum test, with significance levels denoted as follows: *P<0.05, **P<0.01, ***P<0.001, ns: not significant. (D) Discrimination between senescent and non-senescent single cells using PaaSc, CelliD, and GSDensity. Receiver operating characteristic (ROC) curves demonstrate the ability of the three tools to discriminate between senescent and non-senescent single cells in three datasets (GSE102090, GSE119807, and GSE115301). (E) Identification of pathways associated with cell senescence. An association analysis between the activities of the senescence signature and hallmark pathways from MSigDB revealed pathways positively correlated with senescence across five cell types (CD8+T cells, CD4+T cells, B cells, NK cells, and monocytes/macrophages).

## Pathway activity scoring in the presence of batch effects

Batch effects are pervasive across all types of high-throughput biological platforms, including scRNA-seq. To address this challenge, several methods have been developed to correct batch effects in cell embedding spaces, such as Harmony [29], Scanorama [30], and scVI [31], which facilitate accurate cell annotation and visualization. However, while these correction methods focus on cellular embeddings, pathway scoring tools—which operate at the gene expression level—have received limited evaluation for their robustness to batch effects. To assess this robustness, we evaluated PaaSc on the public dataset GSE96583 (Fig 6A), which profiled PBMCs from systemic lupus erythematosus (SLE) patients under two conditions: control (ctrl) and interferon (IFN) stimulation (stim). This dataset exhibits clear batch effects, with cells from the two conditions forming distinct clusters in the embedding space (Fig 6B).

We first examined the performance of different scoring methods on B cell-specific gene sets within B cells. Both PaaSc and CelliD demonstrated high B cell specificity, with the 25th percentile of the rank-normalized scores exceeding 0.95. In contrast, GSdensity showed considerably more variation and inferior performance, with 25th percentile scores below 0.5 in the control condition (Fig 6C). To test whether these tools could inappropriately detect differences due to batch effects alone, we calculated AUC values for distinguishing treated samples from control samples using B cell-specific gene scores. As expected, all three methods yielded AUC values close to 0.5, indicating that cell type-specific gene sets appropriately showed no differential activity between conditions, demonstrating robustness to batch effects. Finally, we compared the ability of the three tools to detect biologically relevant pathway activity by examining IFN pathway scores. Since IFN pathways should be activated in the stimulation condition by experimental design, this serves as a positive control. PaaSc achieved the highest performance, with an AUC of 0.96, substantially outperforming CelliD (0.60) and GSdensity (0.56) (Fig 6D).

## Scoring pathway activities in scATAC-seq data

We further investigated the application of PaaSc to score gene set activities in single-cell ATAC-seq (scATAC-seq) data. Unlike scRNA-seq, which directly measures gene expression, scATAC-seq profiles chromatin accessibility across the genome. Since PaaSc and other computational tools were originally designed for scRNA-seq data, we converted chromatin accessibility signals to gene-level quantifications by calculating UMI counts within promoter regions, which then served as inputs for downstream analysis. To evaluate our approach, we used a publicly available 10x Genomics multiome dataset of human PBMCs that simultaneously measures both DNA accessibility and gene expression in the same cells. Cell types were annotated based solely on gene expression data and served as ground truths for benchmarking. After filtering out cell types with fewer than 100 cells to ensure robust evaluation, we retained 16 distinct cell types for analysis (Fig 7A).

For each cell type, we assessed how effectively cell type-specific signature scores could distinguish target cell types from all others, as measured by the AUC. PaaSc achieved a median AUC of 0.97, outperforming both CelliD (0.93) and GSdensity (0.60) (Fig 7B). When comparing cells correctly predicted by each method, we found that predictions from PaaSc and CelliD largely overlapped, but PaaSc uniquely identified more cells (n=1,343) than CelliD (n=907) (Fig 7C). We next applied PaaSc to explore changes in pathway activity during T cell activation by comparing effector memory

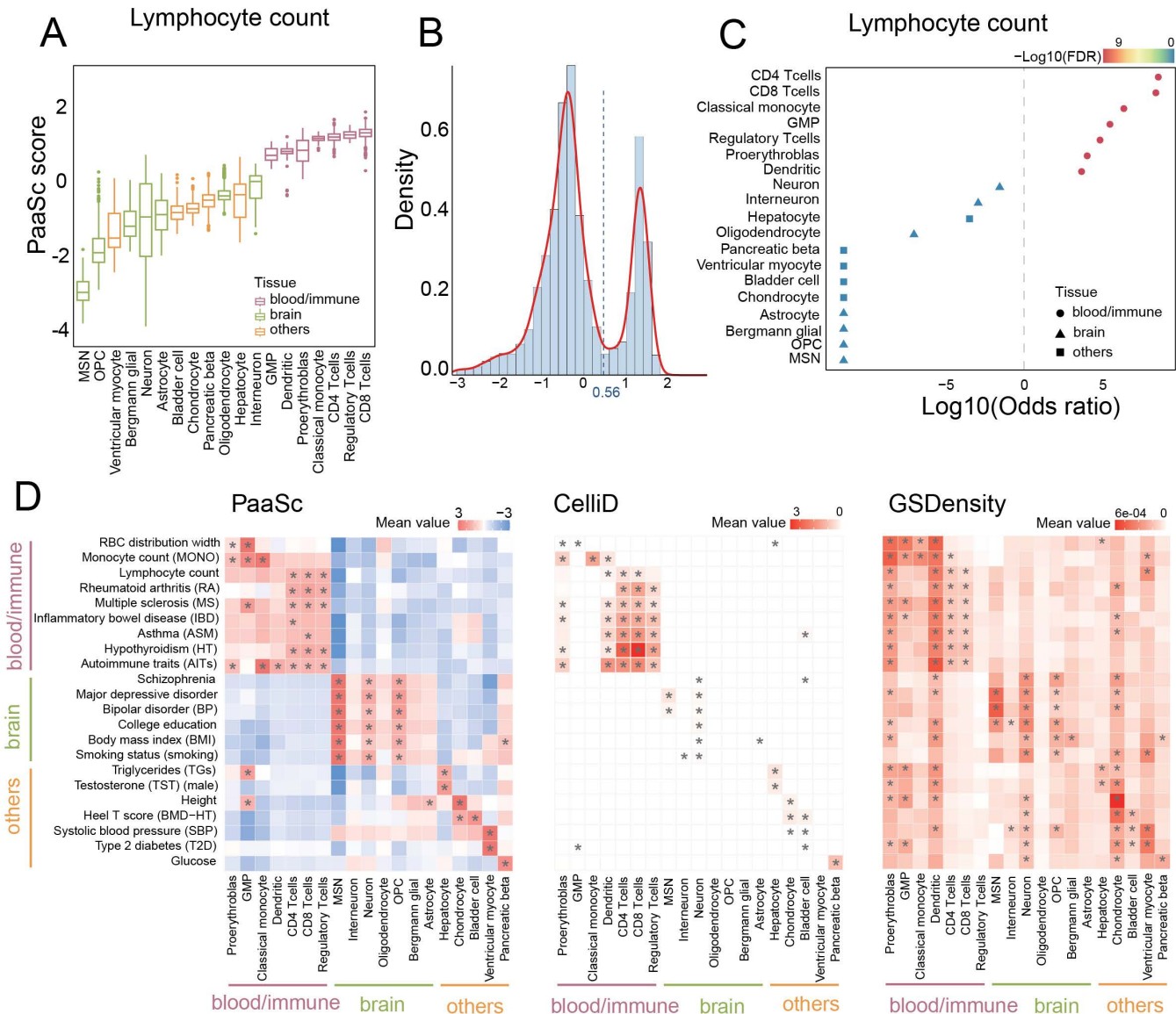

**Fig 5. Identification of GWAS trait-associated cell types using PaaSc.** (A) Enrichment of lymphocyte count-associated genes across 19 sorted cell types. A box plot showing the enrichment scores of lymphocyte count-associated genes calculated by PaaSc. Cell types are categorized into blood, brain, or other categories. (B) Distribution of PaaSc scores for lymphocyte count-associated genes. A histogram illustrating the bimodal distribution of PaaSc scores, enabling the classification of cells into positive and negative states on the basis of the enrichment of lymphocyte count-associated genes. (C) Significance of enrichment of lymphocyte count-associated genes across 19 sorted cell types. Positive cells were defined using the cutoff established in (B). A one-sided Fisher's exact test was performed, and the log10-transformed odds ratio and the negative log10-transformed false discovery rate (–log10 FDR) are shown. (D) Comparison of PaaSc, CelliD, and GSDensity in identifying GWAS trait-associated cell types. Positive cells were defined as the top 5% of cells with the highest scores for each method. Statistical significance was assessed using a one-sided Fisher's exact test, with significant results marked by an asterisk (*).

T cells (CD8 + TEM) with naïve T cells (Fig 7D). Immune-related pathways, including allograft rejection and IL2-STAT5 signaling, as well as proliferation-related pathways, such as E2F targets and the G2M checkpoint, were significantly activated (FDR < 5%). Consistent with previous findings, the WNT and Notch pathways were repressed during T cell activation [32,33].

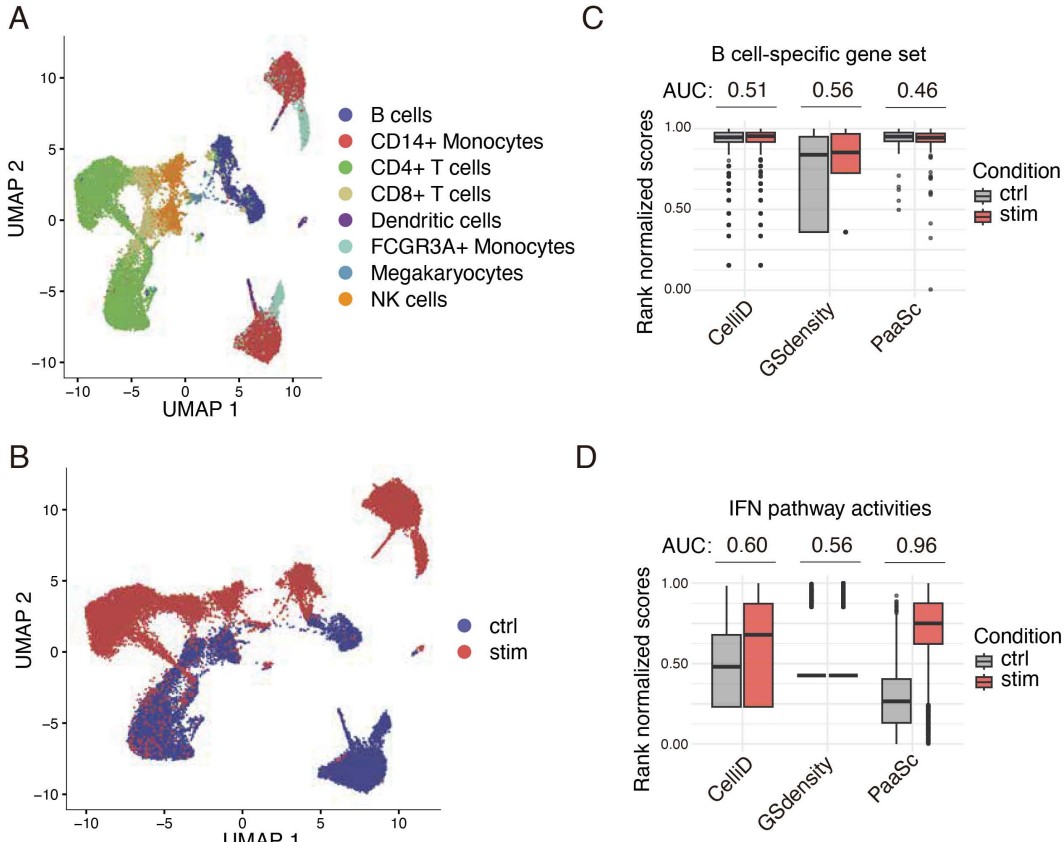

**Fig 6. Scoring pathway activities in the presence of batch effects.** The analysis used the GSE96583 dataset, in which peripheral blood mononuclear cells (PBMCs) from systemic lupus erythematosus (SLE) patients under two biological conditions were profiled: control (ctrl) and interferon (IFN) stimulation (stim). (A, B) Dimensional reduction plots showing cell clustering grouped by cell type (A) and biological condition (B). (C) Evaluation of three pathway scoring methods (PaaSc, GSDensity, and CelliD) using the B cell activation pathway from Gene Ontology. Pathway scores were calculated for each method and normalized by relative rank to values between 0 and 1 (where 1 represents the highest score). The analysis focused on scores from B cells, and receiver operating characteristic (ROC) scores were calculated to assess how effectively each method distinguished between biological conditions on the basis of gene set scores. (D) Interferon pathway activity was calculated using the same approach as in (C), with scores compared between control and stimulated conditions across all cell types.

## Identification of spatially relevant signatures in the human and mouse brain

Building upon previous research that utilized mutual information (MI) for pathway enrichment analysis, we developed a spatial mutual information (SMI) approach to identify spatially relevant pathways (Methods). Pathways were classified as spatially relevant when their activities exhibited non-random distributions across the 2D tissue section. We applied our PaaSc framework to a 10x Genomics Visium dataset of mouse brain sagittal sections [34], which were divided into posterior and anterior regions. Region-specific markers were defined using previously published scRNA-seq data from the mouse brain [35]. As expected, the signatures of the cortex layers were enriched in the corresponding region (Fig 8A). For example, L2/L3 markers are activated mainly in layers 2 and 3 of the mouse cortex (SMI = 0.82, permutation p < 0.0001). Similarly, the cell markers L4, L5 and L6 presented higher gene set activity in the corresponding domain. Vasoactive intestinal polypeptide (VIP) is a neuropeptide found in the cerebral cortex that plays a role in neuronal activity, blood flow, and energy metabolism. Gene markers of VIP neurons are activated pervasively in the mouse cortex and the main olfactory bulb (SMI = 1.04, permutation p < 0.0001) (S7 Fig). In addition, markers of vascular leptomeningeal cells (VLMCs) are

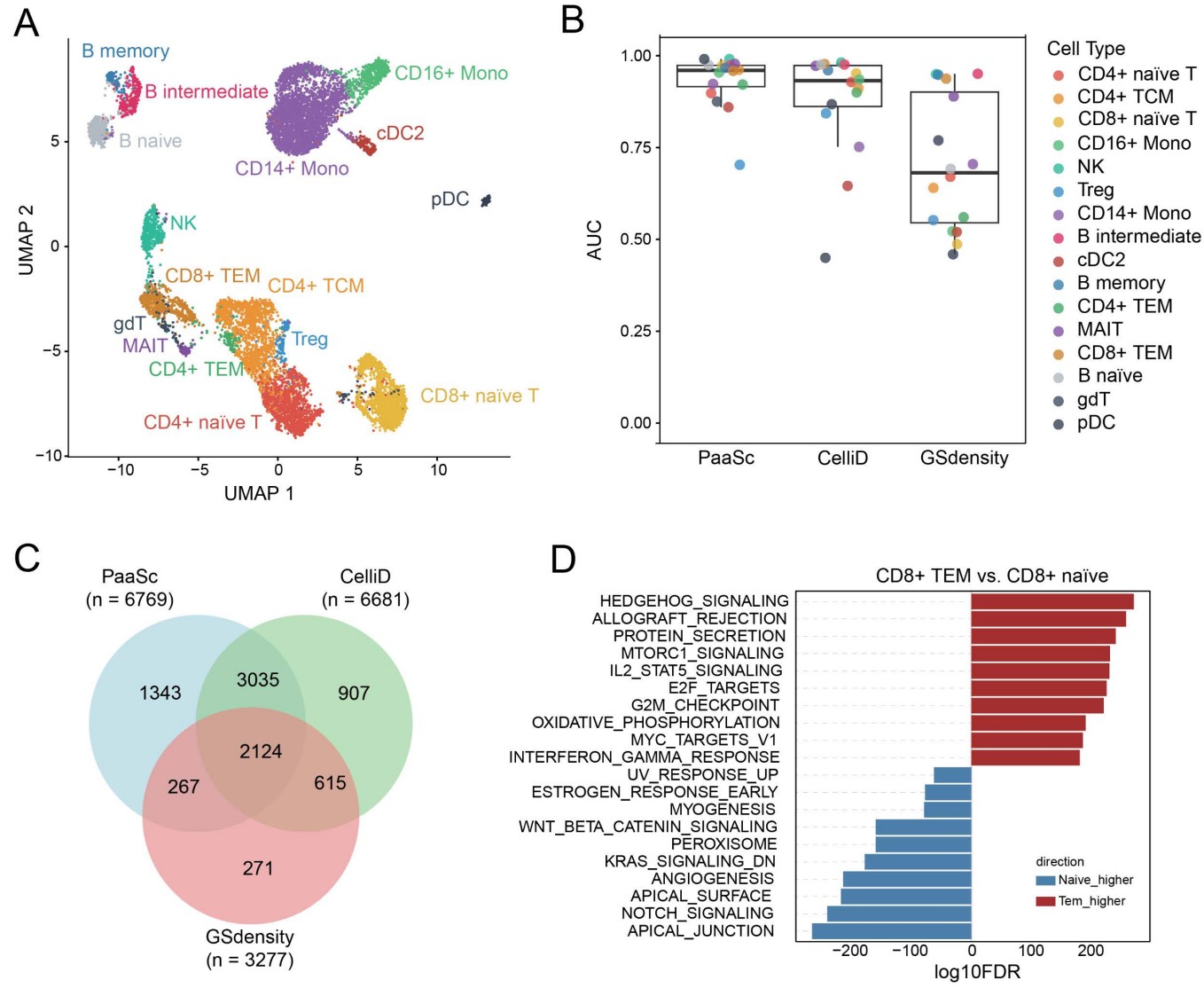

**Fig 7. Evaluation of the performance of PaaSc in evaluating pathway activity using scATAC-seq data.** (A) UMAP visualization of human PBMC single cells from a publicly available 10x Genomics multiome dataset, in which DNA accessibility and gene expression were simultaneously measured. Cell type annotations were transferred from an existing PBMC reference dataset using Seurat package tools [43], utilizing only gene expression information (see Methods). (B) Comparison of cell type discrimination performance between PaaSc, CelliD, and GSDensity using cell type-specific gene sets. Box plots show ROC AUC values representing each method's ability to distinguish target cell types from all other cell types. (C) Venn diagram illustrating the overlap of correctly predicted cells among the three methods. For each method, individual cells were assigned to the cell type with the highest score using the corresponding cell type-specific gene set. (D) Differential pathway activity analysis between CD8+TEM cells and CD8+naïve cells.

activated in the outer layers of the mouse cortex as well as in boundaries of different compartments that are enriched in the extracellular matrix [36] (SMI = 0.48, permutation p < 0.0001).

To quantitatively compare the performance of gene set scoring tools on spatial transcriptomics data, we utilized a public human brain dataset with annotated cortical layers 1–6 and white matter (WM) (Fig 8B). In low-dimensional space, anatomically proximal cells showed spatial clustering patterns, where cells located close to each other in the brain tissue tended to cluster together (Fig 8C). When evaluating performance using domain-specific markers, PaaSc

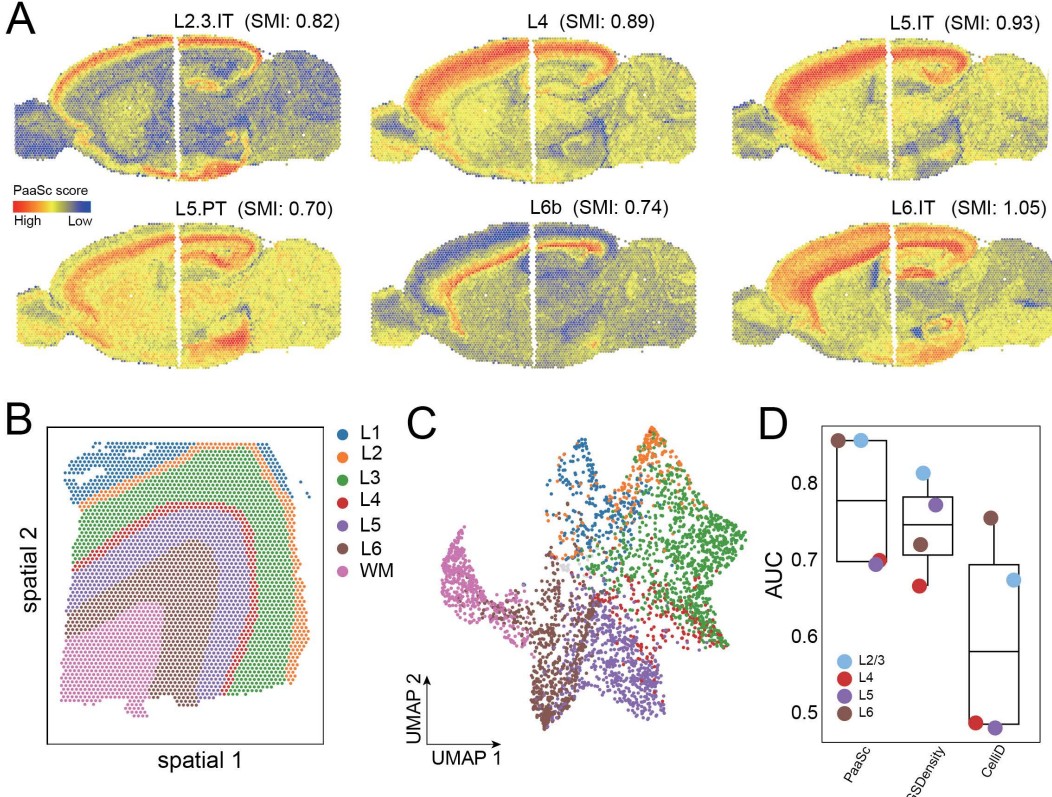

**Fig 8. Identification of spatially relevant signatures using PaaSc in human and mouse brain datasets.** (A) Spatial plots of mouse brain sagittal sections showing PaaSc-calculated activity scores for region-specific gene markers. The color intensity represents the pathway activity score, with higher scores indicating stronger activation. Spatial mutual information (SMI) values are indicated for each gene set. (B) Spatial plot of human brain tissue showing anatomically annotated regions, including cortical layers 1–6 and white matter (WM). (C) UMAP embedding of human brain spatial transcriptomics data demonstrating that anatomically proximal cells cluster together in low-dimensional space. (D) Box plot comparing the performance of PaaSc, GSDensity, and CelliD in distinguishing brain regions using domain-specific markers.

demonstrated superior performance, with a median AUC of 0.78, outperforming both GSDensity (0.75) and CelliD (0.58) (Fig 8D).

## Identification of spatially relevant pathways in lung cancer

We next evaluated the performance of PaaSc on a spatial transcriptomics dataset of human lung cancer acquired using CoxMxSMI [37] (Fig 9A). Cell annotations were defined according to the original study, which identified tumor cells, epithelial cells, endothelial cells, and 12 immune cell types (Fig 9B). To assess the accuracy of cell type identification, we used predefined cell type-specific markers to test how well gene set scores could distinguish each cell type from all others. PaaSc achieved the best performance, with a median AUC of 0.97, followed by CelliD (median AUC = 0.95), although this difference was not statistically significant (paired t test p = 0.06). In contrast, GSDensity showed poor performance, achieving AUCs below 0.8 for all cell types, with a median of 0.66 (Fig 9C).

We then assessed the spatial relevance of both cell types and biological pathways. All cell types demonstrated statistically significant spatial clustering (permutation p < 0.0001, FDR < 5%), with tumor cells exhibiting the greatest spatial relevance (SMI = 0.29). With respect to pathway analysis, the most spatially relevant pathways were predominantly activated in tumor cells, as evidenced by positive correlations between their pathway activity scores and tumor cell gene set scores

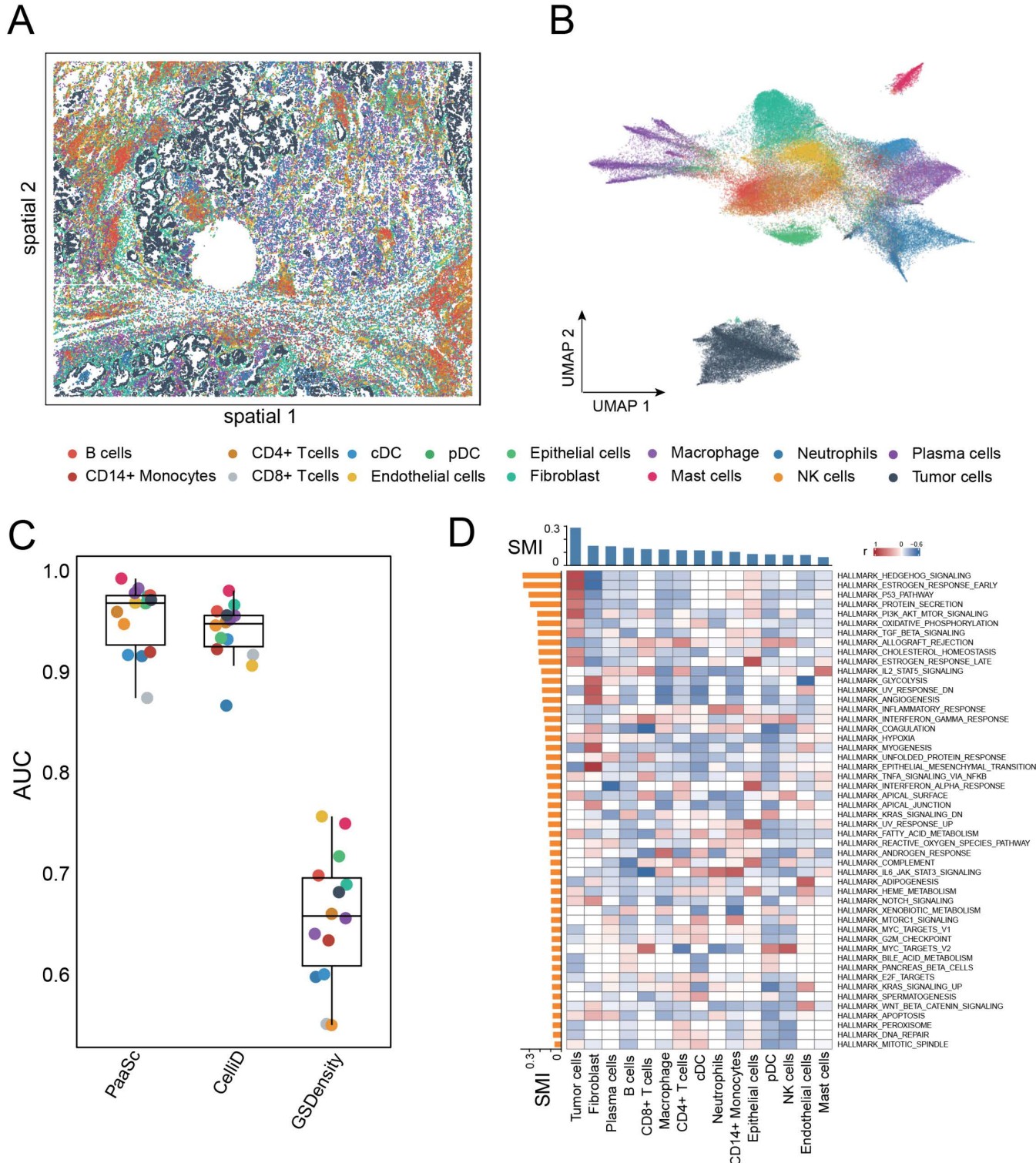

**Fig 9. Performance evaluation of PaaSc on spatial transcriptomics data from human lung cancer.** (A) Spatial plot showing the tissue architecture of human lung cancer samples analyzed using CosMx SMI technology. (B) UMAP embedding showing the distribution of annotated cell types, including tumor cells, epithelial cells, endothelial cells, and 12 immune cell populations. (C) Box plot comparing the performance of PaaSc, CelliD, and GSDensity

in distinguishing individual cell types using cell type-specific markers. Each point represents the AUC value for a specific cell type, with median values indicated. (D) Heatmap showing the correlation between pathway activity scores (rows) and cell type gene set scores (columns). The color intensity indicates the strength of positive (red) or negative (blue) correlations.

(Fig 9D). These pathways included HEDGEHOG SIGNALING, P53 PATHWAY, PI3K AKT MTOR SIGNALING, OXIDATIVE PHOSPHORYLATION, and TGF BETA SIGNALING. Notably, ESTROGEN RESPONSE EARLY was activated in both normal epithelial cells and tumor cells, whereas ALLOGRAFT REJECTION, which also showed high spatial dependence, was primarily activated in immune cell types.

## Discussion

Recent technological advances have enabled the transcriptomic profiling of tens of thousands of cells at single-cell resolution, generating vast datasets that are cataloged in databases such as CELL×GENE [38]. However, obtaining meaningful biological insights from these complex datasets remains a significant challenge. To address this limitation, computational methods have been developed to infer pathway activities at the single-cell level. These methods generate activity scores that can directly capture cellular heterogeneity, making them particularly valuable for investigating developmental trajectories, cellular plasticity, and dynamic response states. In this study, we present PaaSc, a novel computational framework for inferring pathway activities at single-cell resolution. Through comprehensive evaluation, compared with existing state-of-the-art tools, PaaSc demonstrated superior performance in terms of cell type-specific gene set scoring and pathway activity inference.

The better performance of PaaSc is attributed mainly to two aspects. It utilizes MCA for matrix factorization, which projects both genes and cells to the same high-dimensional space, making it easy to identify the nearest genes for each cell. Most importantly, PaaSc estimates the relevance of each factor and selects only the most important factors for pathway activity inference. However, this challenge is far from being solved, as described below.

A major challenge in evaluating pathway scoring methods is the lack of benchmark datasets. Ideally, benchmark data would contain known pathway activity measurements at single-cell resolution, but such datasets are rare. As an alternative strategy, many studies use preannotated cell types as ground truths and test how well cell type-specific gene set scores can distinguish target cells from other cell types. Both CelliD and GSDensity employed this benchmarking approach to evaluate their performance. However, cell type-specific gene sets primarily define cellular identity rather than pathway activity, making it unclear how well these tools perform when scoring biologically relevant pathways. To address this limitation, we included a dataset in which cells were profiled before and after interferon stimulation, providing a more biologically relevant benchmark. Our analysis revealed that while both CelliD and GSDensity accurately identified B cells using B cell-specific markers, they performed poorly when distinguishing between the same cell type under different treatment conditions.

Another major challenge is the incomplete definition of signaling pathways in existing databases. For convenience, most databases define pathways simply as gene sets [5]. However, these gene sets typically contain pathway members (such as receptors and kinases) rather than downstream target genes, making them unsuitable for accurately scoring signaling pathway activity [39]. A more promising approach involves the use of gene expression changes to quantitatively define signaling pathway perturbations, although this method has been limited primarily to cytokine signaling pathways [40]. Fortunately, advances in Perturb-seq—a high-throughput screening technique that combines CRISPR-based gene editing with single-cell RNA sequencing—offer the potential to accurately model a much broader range of signaling pathways [41].

## Methods

### Processing of bulk RNA-seq data

Raw sequence files were obtained from the NCBI Gene Expression Omnibus (GEO) database in Sequence Read Archive (SRA) format. These files were converted to FASTQ format using SRA Toolkit (version 2.9.1). Quality assessment of the

raw sequences was performed using FastQC (version 0.11.8). Subsequently, adapter sequences and low-quality reads were removed using Trim Galore! (version 0.6.2), with default parameters applied in either paired-end or single-end mode according to the sequencing protocol.

Reads were aligned to the human genome hg38 or mouse genome mm10 using STAR [42] with the following parameters: "--quantMode GeneCounts --twopassMode Basic". Differential gene expression was performed using DESeq2, and the resulting test statistics were used to rank genes for gene-set enrichment analysis using GSEA.

### Analysis of single-cell RNA-seq data

Data processing and visualization for most single-cell datasets were performed using Seurat (version 4.3.0) [43] unless otherwise specified. Cell type annotations, which were preexisting in the dataset, were included at the time of download. PaaSc utilizes raw counts as input, whereas the other benchmarking methods were implemented using default parameters and specified input types as described in their respective studies. Pseudotime analysis was performed using the DDRTree algorithm implemented in Monocle2 (version 2.22.0) [44], with trajectories defined by default parameters based on differentially expressed genes between senescent and non-senescent cells, as reported by Teo et al. [20].

Region-specific markers were obtained using the FindMarkers function in Seurat with default parameters, and the top 200 genes were selected on the basis of adjusted p-values. Spatial transcriptomics data were processed and visualized using Scanpy (version 1.10.4), with the Allen Brain Atlas serving as the reference for anatomical regions.

### Processing of 10x multiomics data

To evaluate the performance of PaaSc on scATAC-seq data, we utilized a publicly available 10x Genomics multiome dataset that simultaneously profiles chromatin accessibility (ATAC-seq) and gene expression (RNA-seq) within the same individual cells. The dataset was processed using Signac and Seurat (version 4.3.0) [43]. Quality control filtering was applied to retain high-quality cells on the basis of the following criteria: nucleosome_signal < 2 and TSS.enrichment > 1. For the chromatin accessibility data, gene activity scores were calculated as the total counts within promoter regions (up to 2000 bp upstream) using the 'GeneActivity' function from the Signac R package.

RNA UMI counts were normalized using 'sctransform' [45] implemented in the Seurat package. Cell type annotations were established using the RNA expression profiles as the ground truth. Specifically, cell types were defined by integration with an annotated PBMC reference dataset using the FindTransferAnchors function in Seurat [43].

### MCA method for co-embedding cells and genes into low-dimensional space

Let $\boldsymbol{X} = (x_{ij})$ denote a scRNA-seq expression matrix, where rows represent N cells ($c_1, c_2, \cdots, c_N$) and columns represent M genes ($g_1, g_2, \cdots, g_M$). Each matrix element $x_{ij}$ corresponds to the expression level of gene $j$ in cell $i$. A fuzzy-coded indicator matrix $\boldsymbol{Y} = y_{ij}^{+,-}$ was subsequently constructed, where rows represent N cells ($c_1, c_2, \cdots, c_N$) and columns represent 2M gene states ($g_1^+, g_1^-, g_2^+, g_2^-, \cdots, g_M^+, g_M^-$). The elements of matrix $\boldsymbol{Y}$ are derived by applying min–max normalization to scale the expression values of each gene across all cells to the interval [0,1], which can be expressed as follows:

$$y_{ij}^+ = \frac{x_{ij} - min(x_{\cdot j})}{max(x_{\cdot j}) - min(x_{\cdot j})}, y_{ij}^- = 1 - y_{ij}^+$$

(1)

where $x_{\cdot j}$ is the column j vector of matrix $\boldsymbol{X}$. The superscripts + and - represent two distinct categorical states for each gene. $\boldsymbol{Y}$ is normalized to obtain a probability matrix $\boldsymbol{P}$:

$$P = \frac{Y}{sum(Y)} = \frac{Y}{\sum_i \sum_j y_{ij}} = \frac{Y}{MN}$$

(2)

The matrix of standardized relative frequencies $\boldsymbol{Z}$ is calculated by $\boldsymbol{Z} = \boldsymbol{D}_r^{-1/2}\boldsymbol{P}\boldsymbol{D}_c^{-1/2}$, where $\boldsymbol{D}_r$ and $\boldsymbol{D}_c$ are diagonal matrices with diagonal values equal to the row sums ($\boldsymbol{P}\mathbf{1}_{2M}$) and column sums ($\mathbf{1}_N^T\boldsymbol{P}$), respectively, and $\mathbf{1}_n$ is a column vector containing n elements of 1. Then, singular value decomposition (SVD) is performed on $\boldsymbol{Z}$ such that $\boldsymbol{Z} = \boldsymbol{U}\boldsymbol{D}_\alpha\boldsymbol{V}^T$, where both $\boldsymbol{U}$ and $\boldsymbol{V}$ are orthogonal, and $\boldsymbol{D}_\alpha$ is a diagonal matrix with singular values. The coordinates of the cells and genes are $\boldsymbol{C} = \boldsymbol{D}_r^{-1/2}\boldsymbol{U}$ and $\boldsymbol{G} = \boldsymbol{D}_c^{-1/2}\boldsymbol{Z}^T\boldsymbol{U}$, respectively.

We used an MCA version implemented in CelliD [11] that retains the coordinates corresponding to the positive gene states ($g_1^+, g_2^+, \cdots, g_M^+$), which represent active gene expression patterns. For downstream analysis, the top k dimensions ($k \in [20, 50]$) that explained the most variation were retained.

## Selection of dimensions associated with the pathway of interest

Through MCA decomposition, cells and genes are co-embedded in a *K*-dimensional space. Let $\boldsymbol{C} = (C_1, C_2, \cdots, C_N)^T \in \mathbb{R}^{N \times K}$ denote the cell coordinate matrix and $\boldsymbol{G} = (G_1, G_2, \cdots, G_K)^T \in \mathbb{R}^{M \times K}$ denote the gene coordinate matrix. To identify dimensions associated with the pathway of interest, the gene coordinate vector $\boldsymbol{G}_i = (g_{1i}^+, g_{2i}^+, \cdots, g_{Mi}^+)^T$ was treated as the dependent variable for each MCA dimension. The pathway occurrence matrix $\boldsymbol{S} = (S_1, S_2)^T = (s_{ij})$ served as the independent variable, where $s_{i1}$ equals 1 if the *i*-th gene belongs to the target gene set and 0 otherwise. Additionally, $s_{i2}$ represents the frequency of gene occurrence across all possible pathways.

A linear regression model is then applied to establish the following relationship:

$$G_i = \alpha\mathbf{1} + S\beta + e, \ E(e) = 0, \ Cov(e) = \sigma^2 I, \tag{3}$$

where $\boldsymbol{\beta} = (\beta_1, \beta_2)^T$ is the vector of unknown parameters, $\boldsymbol{e}$ is the vector of random errors, satisfying the Gauss–Markov assumptions, and $\alpha$ is the intercept term. Least squares estimation is used to estimate the regression parameters. The least squares estimate of the regression coefficient is given by:

$$\hat{\beta}(k) = (S'S)^{-1} S'^{G_i}. \tag{4}$$

A t test was subsequently performed for each regression coefficient to evaluate its significance. The t-statistics were computed as follows:

$$t_{ij} = \frac{\hat{\beta}_i}{SE(\hat{\beta}_i)}, \tag{5}$$

where $SE(\hat{\beta}_i)$ is the standard error of the estimated coefficient. The corresponding p-value was calculated as follows:

$$p_{ij} = P\left(t_{M-3} > |t_{ij}|\right), \tag{6}$$

where $t_{M-3}$ follows a t-distribution with $M-3$ degrees of freedom.

## Calculation of gene pathway activity at the single-cell level

Linear regression analysis was performed to assess the association between gene coordinates in each MCA dimension and target genes. Only dimensions with statistical significance ($p < 0.05$) are retained. The test statistics $t_j$ are utilized as weights for the cell coordinates, defined as follows:

$$W_{ij} = \begin{cases} t_{ij}, & \text{if } p < 0.05 \\ 0, & \text{if } p > 0.05 \end{cases} \tag{7}$$

Weights are subsequently normalized to ensure comparability across dimensions. The normalized weight matrix is defined as $\boldsymbol{W} = (W_1, W_2) = (w_{ij}^*) = (\frac{w_{ij}}{\sum_i w_{ij}})$. The target gene scores and background gene set scores at the single-cell level are obtained by computing the sum of weighted cell coordinates. The target gene scores are standardized using z-score normalization, with the background gene set scores constituting the basis for the standardization process: $score = \frac{\boldsymbol{CW}_1 - (mean(\boldsymbol{CW}_2))\mathbf{1}_N}{sd(\boldsymbol{CW}_2)}$. A higher score corresponds to greater activity within the gene pathway at the single-cell level.

## Definition and discrete approximation of spatial mutual information

Spatial mutual information (SMI) extends the definition of mutual information. The SMI between location $(X, Y)$ and variable $Z$ is defined as follows:

$$I((X, Y); Z) = \sum_{x,y,z} p(x, y, z) log \frac{p(x, y, z)}{p(x, y)p(z)} \tag{8}$$

where $p(x, y, z)$ is the joint probability distribution, $p(x, y)$ is the marginal probability distribution of locations $(X, Y)$, and $p(z)$ is the marginal probability distribution of the variable $Z$.

For computational purposes, we discretize the continuous space into several bins. For the joint probability distribution and marginal probability distribution:

$$p(x, y, z) = \frac{n(x, y, z)}{N} \tag{9}$$

$$p(x, y) = \frac{\sum_k n(x, y, z_k)}{N} \tag{10}$$

$$p(z) = \frac{\sum_{i,j} n(x_i, y_j, z)}{N} \tag{11}$$

where $N$ is the total number of observations, $n(x, y, z)$ is the count in each $(x, y, z)$ bin, $\sum_k n(x, y, z_k)$ is the sum of the counts in all $z$ bins, and $\sum_{i,j} n(x_i, y_i, z)$ is the sum of the counts in all $(x, y)$ bins.

## Supporting information

**S1 Fig. Pseudocode of PaaSc.**
(TIFF)

**S2 Fig. Schematic of the PaaSc workflow using the human PBMC3k dataset.** (A) Bar plot depicting the significance of each dimension (n = 50) as determined by linear regression analysis. (B) UMAP visualization showing nine distinct PBMC populations: B cells, naïve CD4 + T cells, memory CD4 + T cells, CD8 + T cells, natural killer (NK) cells, plasmacytoid dendritic cells (pDCs), conventional dendritic cells (DCs), CD14 + monocytes, CD16 + monocytes, and FCGR3A+ monocytes. (C) UMAP projection colored according to B cell activation pathway scores. (D) Histogram illustrating the bimodal distribution pattern of B cell activation pathway scores. (E) UMAP representation with binarized classification labels. (F) Comparative box plot displaying B cell activation pathway scores across different cell subtypes. (G) ROC curve demonstrating the discriminative power of B cell activation pathway scores for distinguishing B cells from non-B cell populations.
(TIFF)

**S3 Fig. Cell type prediction performance of PaaSc using REAP-seq data.** (A) Box plots comparing the average expression levels of cell type-specific markers, with target cell populations annotated in red. (B) Heatmap representation of the confusion matrix demonstrating the classification accuracy of PaaSc using cell type-specific markers. The color gradient intensity corresponds to the number of cells within each category.
(TIFF)

**S4 Fig. Benchmarking using B cell-specific markers.** Box plot comparing the performance of PaaSc and 7 competing tools in distinguishing B cells from other cell populations, as measured by AUC scores based on B cell-specific markers.
(TIFF)

**S5 Fig. Benchmarking of tools for cell senescence signature scoring.** (A–D) Pathway activity score distributions calculated by PaaSc, CelliD, and GSDensity across four datasets. (E) Correlations between the median activity score and number of passages. (F) Identification of pathways associated with cell senescence by GSDensity.
(TIFF)

**S6 Fig. Identification of GWAS trait-associated cell types.** (A) Box plots showing the enrichment scores of lymphocyte count-associated genes calculated by CelliD and GSDensity. (B) Distribution of pathway scores calculated by PaaSc, CelliD, and GSDensity. Dotted lines indicate the 95% quantile.
(TIFF)

**S7 Fig.** Spatial distribution of domain-specific gene sets in the mouse brain.
(TIFF)

**S1 Table. Cancer scRNA-seq datasets from the TISCH database.**
(XLSX)

**S2 Table. Cell type-specific gene-set scores by PaaSc and other methods.**
(XLSX)

**S3 Table. Bulk RNA-seq datasets for senescence signature validation.**
(XLSX)

## Acknowledgments

We are grateful for the high-performance computing center of the Center for Excellence in Molecular Cell Science (CEMCS), CAS, for its support in data processing.

## Author contributions

**Conceptualization:** Jiantao Shi.

**Data curation:** Xiqi Liao, Yan Feng.

**Formal analysis:** Xiqi Liao, Yan Feng.

**Funding acquisition:** Jiantao Shi.

**Investigation:** Xiqi Liao, Yan Feng.

**Methodology:** Xiqi Liao, Yuyang Hong, Henghui Li, Jiantao Shi.

**Project administration:** Jiantao Shi.

**Resources:** Hai Fang, Jiantao Shi.

**Software:** Yuyang Hong, Henghui Li.

**Supervision:** Hai Fang, Jiantao Shi.

**Visualization:** Xiqi Liao, Yuyang Hong, Yan Feng.

**Writing – original draft:** Xiqi Liao, Yuyang Hong, Yan Feng, Henghui Li, Hai Fang, Jiantao Shi.

**Writing – review & editing:** Xiqi Liao, Yuyang Hong, Yan Feng, Henghui Li, Hai Fang, Jiantao Shi.

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
