## [Decision Letter · Decision Letter 0]

30 Jun 2025

Inferring pathway activities for single-cell and spatial transcriptomics data with PaaSc

PLOS Computational Biology

Dear Dr. Shi,

Thank you for submitting your manuscript to PLOS Computational Biology. After careful consideration, we feel that it has merit but does not fully meet PLOS Computational Biology's publication criteria as it currently stands. Therefore, we invite you to submit a revised version of the manuscript that addresses the points raised during the review process.

Please submit your revised manuscript within 60 days Aug 30 2025 11:59PM. If you will need more time than this to complete your revisions, please reply to this message or contact the journal office at ploscompbiol@plos.org. Please include the following items when submitting your revised manuscript:

We look forward to receiving your revised manuscript.

Kind regards,

Wei Lan

Academic Editor

PLOS Computational Biology

Ilya Ioshikhes

Section Editor

PLOS Computational Biology

**Journal Requirements:**

4) We notice that your supplementary Figures are included in the manuscript file. Please remove them and upload them with the file type 'Supporting Information'. Please ensure that each Supporting Information file has a legend listed in the manuscript after the references list. 

Potential Copyright Issues: 

i) Figure 6. Please confirm whether you drew the images / clip-art within the figure panels by hand. If you did not draw the images, please provide (a) a link to the source of the images or icons and their license / terms of use; or (b) written permission from the copyright holder to publish the images or icons under our CC BY 4.0 license. Alternatively, you may replace the images with open source alternatives. See these open source resources you may use to replace images / clip-art:

6) We note that your Data Availability Statement is currently as follows: "All relevant data are within the manuscript and its Supporting Information files.". Please confirm at this time whether or not your submission contains all raw data required to replicate the results of your study. Authors must share the “minimal data set” for their submission. PLOS defines the minimal data set to consist of the data required to replicate all study findings reported in the article, as well as related metadata and methods (https://journals.plos.org/plosone/s/data-availability#loc-minimal-data-set-definition). 

7) Please amend your detailed Financial Disclosure statement. This is published with the article. It must therefore be completed in full sentences and contain the exact wording you wish to be published. 

8) Please send a completed 'Competing Interests' statement, including any COIs declared by your co-authors. If you have no competing interests to declare, please state "The authors have declared that no competing interests exist". Otherwise please declare all competing interests beginning with the statement "I have read the journal's policy and the authors of this manuscript have the following competing interests"

**Reviewers' comments:**

Reviewer's Responses to Questions

**Comments to the Authors:**

Reviewer #1: In the manuscript, a method called the PaaSc is proposed for inferring pathway activities for single-cell data. Benchmark comparisons with some other existing methods are done on several publicly available datasets. However, the manuscript is written in an unintelligible way, and I found it hard to comprehend the novelty of the method accurately and understand why it is better than other benchmark methods. The manuscript should be thoroughly reorganized and rewritten in a clearer and more understandable way before I can better evaluate the strengths of the proposed method.

1. The introduction of the implementation of the proposed method is confusing. The authors should present a flowchart or an Algorithm box to show how the PaaSC method works.

2. The organization of the manuscript is somewhat confusing. The captions should align with the figures, and I found it very difficult to go back and forth to check the figures and then jump back to the captions for illustration.

3. In Fig. 2, the AUC scores from different methods are compared. Why does the AUC score matter? I am not fully convinced that the proposed method performs better than the other methods. Also, it seems that GSdensity and CelliD yield a similar AUC score to the proposed PaaSC method.

4. Continuing with 3, can the authors provide some theoretical justifications on why the proposed method performs better than the benchmarks? If some insightful understanding could be provided, readers would have a better understanding of the merits of the proposed method.

5. There are some abbreviations that make it harder to understand the manuscript. E.g., FDR, AUC, SMI... I would suggest making a list of notations at the beginning.

6. Introductions on distinct datasets and their underlying biological significance will be helpful for the readers to understand better why to use those datasets.

7. How are the thresholds for p and FDR determined? I noticed that different thresholds are used across different examples. Some are 0.001 while others are 0.05 or 0.0001.

Reviewer #2: The authors developed a computational method, named PaaSc, for inferring pathway activities at single cell level. The results on diverse benchmark datasets demonstrated superior performance over existing SOTA methods in multiple applications. This work has made certain improvements to the published methods. But there are several concerns should be addressed:

(1)The term "pathway activities" is mentioned in the title, yet it is not directly reflected in the subtitles. It gives the impression of being a compilation of experimental results. Is it necessary to include a paragraph elaborating the association between “pathway” and the experiments?

(2)The AUC metric in line 36 and line 90 is the same term? If the same, please maintain consistency in using the full term.

(3)In line 123, the name of the database should be TISCH.

(4)In line 177, “IL 16” does not appear in Figure 4E.

(5)In line 190, there should not be a space between “CD ” and “4”.

(6)In line 206, the middle panel should be the right panel.

(7)In line 210, the SMI approach is not described in detail in the Method section, and it's also unclear what the value of SMI represents.

(8)In line 313, what dose the “P12M” mean?

Reviewer #3: Manuscript Title: Inferring pathway activities for single-cell and spatial transciptomics data with PaaSC

Summary of the Study

This manuscript addresses a major problem that is being observed across science disciplines: “translating high-dimensional data into functional pathway insights”. This manuscript really begins to address a broader problem the scientific community faces in that we generate mass quantities of data but do not fully utilize the data for discovery due to limitations in tools easily accessible or even availablity. The authors developed PaaSC to address this problem with highlighting the novelty of using biologically informative components through linear regression relative to other similar models (CelliD, GSDensity) which they thoroughly benchmarked against. Through a series of rigorous computational benchmarks, the authors demonstrate that PaaSC outperforms other similar platforms in MOST scenarios. There is a need to really interrogate the processes on how data are processed for analysis. This is one area that biologist overlook not fully understanding the limitations of computational approaches in data analysis. The authors clearly understand this and have taken steps towards improving this. While the overall topic is of interest and relevant to the journal, the manuscript suffers from some minor technical, editorial, and conceptual weaknesses in its current form.

General comments:

• The manuscript was well written with clearly framing the gaps in knowledge, the current state of the field, and where improvements need to be made in data analysis. This was very thoughtfully put together. The authors did very well in presenting a current limitation in analyses that have difficulties reliably identifying subtle and contextual biological changes. There is huge value in understanding subtle changes which goes to understanding cell groundstate or plasticity, an area of research more and more are accepting and going towards in our fundamental understanding of developmental biology.

• The discussion section overall read more like a summary of findings rather than a discussion. I strongly suggest the authors reframe this section, consider what their approach means, where future improvements lie and what are the major next hurdles to conquer (not just PaaSC).

• Reading the text and assessing the data, the language used appears to exaggerate the findings and comes off as overstating the performance. “superior” is a qualitative term and I would suggest changing this to use more quantitative measure. This will overcome that issue. For example, in line 165 the authors have done this and therefore can now use that term because they included the quantitative value.

• Qualitative Overreliance in the Results Section: The manuscript repeatedly uses qualitative descriptors such as “increased,” “reduced,” and “enhanced” without reporting actual fold-changes or percent differences. This weakens the impact of the findings and makes it difficult to evaluate the robustness of the effects. As the authors have done the analyses, they should include them in the narrative of the results section.

• The spatial transcriptomic analysis rationale was strongly built but the outcome and what was delivered was quite underwhelming presented. This is huge area that needs to be highlighted more as there is a massive shift for people using these types of analyses. I strongly recommend the authors expand on this section. Because they spent so much time benchmarking the other data sets and approaches, they undermine themselves by not ding that here as well. Please keep the nice continuity you have built with this section and expand upon it in greater detail.

• Definitions: multiple terms are not defined in the manuscript including PaaSC, REAP-seq, etc….

• The authors discuss and analyze some biological features in Figure 4. I would suggest if they can highlight any novelty they found in these analyses that would have been overlooked by the other benchmarked approaches…this would really strengthen their arguments and convince non-computational individuals about the value of this work.

• Figure and Image Quality Figures general are difficult to follow. This may be a labelling problem. Color coordination patterns are mixed within figures which causes some confusion. A lot of this would be attributed to font size, scaling of the graphs, the tick marks on the axes, the labeling of the graphs. Visualization is extremely important. Figures 2 and 3 use a lot to the same box plots and when looking at this, it is too much data to take in as shown. I would suggest changing the visualization of these box blots to make it clearer what they want to show us.

**Have the authors made all data and (if applicable) computational code underlying the findings in their manuscript fully available?**

Reviewer #1: Yes

Reviewer #2: Yes

Reviewer #3: Yes

PLOS authors have the option to publish the peer review history of their article (what does this mean? ). If published, this will include your full peer review and any attached files.

**Do you want your identity to be public for this peer review?** For information about this choice, including consent withdrawal, please see our Privacy Policy .

Reviewer #1: No

Reviewer #2: No

Reviewer #3: No

**Figure resubmission:**
---

## [Decision Letter · Decision Letter 1]

29 Sep 2025

PCOMPBIOL-D-25-01147R1

Inferring pathway activities for single-cell and spatial transcriptomics data with PaaSc

PLOS Computational Biology

Dear Dr. Shi,

Thank you for submitting your manuscript to PLOS Computational Biology. After careful consideration, we feel that it has merit but does not fully meet PLOS Computational Biology's publication criteria as it currently stands. Therefore, we invite you to submit a revised version of the manuscript that addresses the points raised during the review process.

Please submit your revised manuscript within 30 days Nov 29 2025 11:59PM. If you will need more time than this to complete your revisions, please reply to this message or contact the journal office at ploscompbiol@plos.org. Please include the following items when submitting your revised manuscript:

We look forward to receiving your revised manuscript.

Kind regards,

Wei Lan

Academic Editor

PLOS Computational Biology

Ilya Ioshikhes

Section Editor

PLOS Computational Biology

**Reviewers' comments:**

Reviewer's Responses to Questions

Reviewer #1: The authors have addressed most of my concerns. However, I noticed that new typos and compiling errors were introduced. The mathematical formulas are not compiled corrected starting from Eq. 2. Therefore, I suggest that the authors check the manuscript thoroughly and revise any potential typos and compiling errors again.

Reviewer #3: Thank you for addressing my comments as well as the other reviewers. I am satisfied with the changes you have made. I agree that your manuscript and story have been greatly improved.

**Have the authors made all data and (if applicable) computational code underlying the findings in their manuscript fully available?**

Reviewer #1: Yes

Reviewer #3: Yes

PLOS authors have the option to publish the peer review history of their article (what does this mean? ). If published, this will include your full peer review and any attached files.

**Do you want your identity to be public for this peer review?** For information about this choice, including consent withdrawal, please see our Privacy Policy .

Reviewer #1: No

Reviewer #3: **Yes: ** Eric Rahrmann

**Figure resubmission:**
---

## [Decision Letter · Decision Letter 2]

27 Oct 2025

Dear Dr. Shi,

We are pleased to inform you that your manuscript 'Inferring pathway activity from single-cell and spatial transcriptomics data with PaaSc' has been provisionally accepted for publication in PLOS Computational Biology.

Best regards,

Wei Lan

Academic Editor

PLOS Computational Biology

Ilya Ioshikhes

Section Editor

PLOS Computational Biology

Reviewer's Responses to Questions

**Comments to the Authors:**

Reviewer #1: The authors have fixed the compiling issue and addressed my concerns.

**Have the authors made all data and (if applicable) computational code underlying the findings in their manuscript fully available?**

Reviewer #1: None

PLOS authors have the option to publish the peer review history of their article (what does this mean? ). If published, this will include your full peer review and any attached files.

**Do you want your identity to be public for this peer review?** For information about this choice, including consent withdrawal, please see our Privacy Policy .

Reviewer #1: No

---

## [Editor Report · Acceptance letter]

PCOMPBIOL-D-25-01147R2

Inferring pathway activity from single-cell and spatial transcriptomics data with PaaSc

Dear Dr Shi,

I am pleased to inform you that your manuscript has been formally accepted for publication in PLOS Computational Biology. Your manuscript is now with our production department and you will be notified of the publication date in due course.

With kind regards,

Anita Estes
